# Are Yellow Sticky Cards and Light Traps Effective on Tea Green Leafhoppers and Their Predators in Chinese Tea Plantations?

**DOI:** 10.3390/insects12010014

**Published:** 2020-12-29

**Authors:** Longqing Shi, Haifang He, Guang Yang, Huoshui Huang, Liette Vasseur, Minsheng You

**Affiliations:** 1State Key Laboratory of Ecological Pest Control for Fujian and Taiwan Crops, Institute of Applied Ecology, Fujian Agriculture and Forestry University, Fuzhou 350002, China; shilongqing@faas.cn (L.S.); hehaifang55@outlook.com (H.H.); yxg@fafu.edu.cn (G.Y.); 2Institute of Rice, Fujian Academy of Agricultural Sciences, Fuzhou 350018, China; 3Joint International Research Laboratory of Ecological Pest Control, Ministry of Education, Fuzhou 350002, China; 4College of Plant Protection, Henan Agricultural University, Zhengzhou 450002, China; 5Key Laboratory of Integrated Pest Management for Fujian-Taiwan Crops, Ministry of Agriculture, Fuzhou 350002, China; 6Comprehensive Technology Service Center of Quanzhou Customs, Quanzhou 362000, China; hemenghan@henau.edu.cn; 7Department of Biological Sciences, Brock University, 1812 Sir Isaac Brock Way, St. Catharines, ON L2S 3A1, Canada

**Keywords:** *Empoasca onukii*, tea plantations, yellow sticky cards, light traps

## Abstract

**Simple Summary:**

Leafhoppers are serious insect pests in agriculture across the world. Both nymphs and adults suck the sap of plant shoots and leaves with their piercing–sucking mouthparts causing damage called “hopperburn”. The tea green leafhopper, *Empoasca onukii*, is one of most damaging pests in tea plantations in Asia. In China, yellow sticky cards and light traps are increasingly used to control leafhoppers in tea plantations, especially the tea green leafhopper. Visually, several leafhoppers appear to be captured and killed, however, the real control efficiency and the damage to natural enemies remains unclear. In our study, a 16-week open field experiment with daily weather monitoring was designed to test the responses of tea green leafhopper, parasitoids and spiders to yellow sticky cards and light traps (cover with sticky cards) that used different light colours. An exclosure experiment was also designed to further test the influence of the three light systems (without sticky card) on the same groups of species. The results suggested that light, especially green and white, can be useful as a way to control leafhopper populations without affecting parasitoids and spiders too much.

**Abstract:**

In Chinese tea plantations, yellow sticky cards and light traps are increasingly used to control insect pests, especially the tea green leafhopper *Empoasca onukii*. In this study, a 16-week open-field experiment with daily weather monitoring was designed to test the responses of tea green leafhopper, parasitoids and spiders to yellow sticky cards and three light traps with different wavelengths (covered with sticky cards). An exclosure experiment was also designed to further test the influence of the three light systems (without sticky card) on the same species. The results showed that all three light emitting diode (LED) light traps (white, green and yellow) and yellow sticky cards attracted many more *E. onukii* male adults than females during the course of the open field experiment, with less than 25% of trapped adults being females. Parasitoids and spiders were also attracted by these systems. Weather variables, especially rainfall, influenced the trapping efficiency. In the exclosure experiment, the population of leafhoppers in the yellow sticky card treatment did not decline significantly, but the number of spiders significantly decreased. The green and white light treatments without sticky cards showed a significant control of *E. onukii* and no obvious harm to spiders. These results suggest that yellow sticky cards and light traps have limited capacity to control tea green leafhoppers. However, light, especially green light, may be a promising population control measure for tea green leafhoppers, not as killing agents in the traps, but rather as a behavioral control system.

## 1. Introduction

Tea (*Camellia sinensis*) plantations are considered perennial monocultures with habitat homogeneity and low plant species diversity [1,2,3]. In conventional tea plantations, farmers generally apply a wide range of pesticides to control pest outbreaks. Because tea leaves are rarely washed before being processed, there are concerns regarding human health safety due to the presence of pesticide residues [4,5]. For this reason, several tea growers in China have now converted to organic tea growth.

In China, one of the main tea pests is the tea green leafhopper (*Empoasca onukii*) [6], which can produce 9–17 generations per year, and all life stages can damage the plants [7,8]. The nymphs and adults pierce the young tea leaves or shoots with their mouthparts, and suck the phloem sap, causing serious withering of tea plants, called ‘hopperburn’ [9]. In tea plantations of China, parasitoids and spiders are major predators of leafhoppers and contribute to their population control [10,11]. An increasing number of farmers have realized the importance of these natural predators and tried different alternatives to reduce pesticides and promote their presence.

Various alternative pest control methods have been tested in tea plantations. Among them, yellow sticky cards and light traps are widely used for insect pest control and population monitoring [12,13,14,15]. Yellow sticky cards tend to attract flying pests such as Diptera [16], Coleoptera [17] and Hemiptera [18]. Four pest-control light methods exist: phototaxis to attract pests, host-detection disruption, radiation to kill or suppress, and manipulation of the circadian rhythms [19]. Various types of light traps are available including ultraviolet light and light emitting diode (LED) lights of various wavelengths. They are mostly used to attract and kill nocturnal insects such as moths (Lepidoptera) [20], termites (Isoptera) [21], whiteflies (Hemiptera) [22], and beetles (Coleoptera) [23].

Attraction to light and various colors can be sex-specific, depending on the species. In leafhoppers, such as *Empoasca vitis*, *Empoasca onukii*, and *Scaphoideus titanus* (Hemiptera: Cicadellidae), adult males tend to be more attracted to yellow sticky traps than females [18,24,25]. Male moths *Yponomeuta cagnagella* (Lepidoptera: Yponomeutidae) and *Ligdia adustata* (Lepidoptera: Geometridae) are significantly more attracted to light than females [26]. On the other hand, female chironomids (Diptera) are much more attracted to white light than males [27].

These alternative control techniques may have negative impacts in the ecosystem if they target a broad spectrum of invertebrates, some being beneficial invertebrates such as natural pest control agents. For instance, *Encarsia formosa* (Hymenoptera: Aphelinidae) is an important pest predator that has positive phototaxis and is generally sensitive to the short-wavelength lights [28]. This species is also attracted to yellow sticky cards [29]. The effectiveness of these control techniques can vary depending on environmental factors such as weather conditions.

With the continuous development of Chinese tea, quality and safety have become the primary goals of the tea industry. In China, yellow sticky cards and light traps are increasingly used to control insect pests in tea plantations [12,13,14,15], however, whether these techniques have any effect on predators in tea plantations is still unclear, and the real control efficiency and the damage to natural pest control agents remain unclear. It is therefore important to understand the impacts of such alternative techniques, not only on pests but also on other invertebrates that can be beneficial for pest control. In this study, we designed a field experiment with open plots and exclosures to examine the responses of *E. onukii* population, and communities of parasitoids and spiders in tea plantations where sticky cards and light traps were installed. To acquire a realistic estimate of the population dynamic of male and female tea green leafhopper in tea plantation, a suction machine was used, to sample populations over the trial period [25]. The main goal was to determine whether such techniques might be effective against leafhopper but have negative impacts on natural pest control agents, and if weather conditions could affect their effectiveness.

## 2. Materials and Methods

### 2.1. Study Site

We conducted these experiments in the Hongxing tea plantation located in Quanzhou, Fujian Province, China (approximately 40,000 m^2^, 25°0′15.76″ N, 117°52′0.04″ E; 700–750 m elevation). The tea plants in Hongxing were more than 50 years old and had been organically managed without pesticides and chemical fertilizers for the past 10 years.

### 2.2. Open-Field Experiments

To assess the effects of yellow sticky cards and light traps, five large blocks containing plants of the same height (at least 20 m apart) were selected. In each block, five plots of 3 × 3 m were set up (at least 10 m apart) and were assigned randomly to one of the five treatments: one sticky card, three light treatments and a control with no control method. The first treatment used yellow sticky cards (15 × 20 cm card, with glue on both sides). One card per plot was installed between two rows of tea plants using bamboo sticks at the height of the tea canopy (i.e., approximately 0.5 m).

The three light treatments consisted in the installation of a new light trap design with white (Figure 1a), green (dominant wavelength: 508.0 nm, Figure 1b) or yellow (dominant wavelength: 569.0 nm, Figure 1c) LED lights (the intensity of the three LEDs were approximately 2000 lux). This new system included a modified lamp on which a transparent cylinder-shaped lampshade with a transparent sticky card with the glue facing outside (15 cm height, 40 cm circumference) was installed over the lampshade. Hence, the light traps had the same sticky surface area as yellow sticky cards did. In the center of each light treatment plot, one light trap was installed at the height of the tea canopy (same height as the yellow sticky card) (i.e., three plots, each with one specific light color). Each light was connected to a solar panel for power supply, and a photo-sensitive switch was installed to have the lights on only during darkness. The fifth treatment was the control, where neither lights nor cards were used.

In order to assess the effects of the treatments on leafhoppers, parasitoids and spiders, every day at 7:00 a.m., the sticky cards were collected and replaced. The replaced cards were taken back to the laboratory, the numbers of female and male tea leafhopper adults on the cards were counted, as well as the trapped parasitoids and spiders. All the insects and spiders on the cards were collected and stored in 75% alcohol and identified them under a stereomicroscope. The parasitoids and spiders were identified to the family level. The collected parasitoids (Hymenoptera) were identified based on the differences in antenna, mouth parts, mesonotum, prododeum, wing (vein, pterostigma, etc.), which referred to the book wrote by Rrichards (1985) [30]. The collected spiders (Araneae), were identified mainly according to the morphological characters, the arrangement of ocellus, mouth parts, sternum, appendages, spinnerets and abdomen [31]. The control plots served to understand the normal population dynamics. In this case, a suction machine (electromotor, 16,000 rpm, with a 12-cm-diameter inlet, Figure 2) with the trapping was used to collect and record the numbers of female and male tea leafhopper adults, parasitoids and spiders. Each suction collection was also done at 7:00 a.m., lasted for 2 min, and covered a surface area of 9 m^2^ (surface area expected for daily activities of leafhoppers in tea plantations). The experiment ran from 13 May 2015 to 20 September 2015.

During the study, a weather station (HOBO ware, Bourne, MA, USA) was set near the testing field to record the weather parameters including rainfall (RF), relative humidity (RH), solar radiation (SR), wind speed (WS), and temperature. To accurately match the trapping periods (which were recorded every day at 7:00 a.m.), daily weather data were collected from 7:00 to 6:59 a.m. the following day (Figure 3).

### 2.3. Exclosure Experiment

To assess the control effects of yellow sticky cards and light traps, 25 exclosure cages (3 × 4.5 m, and 2 m in height; Figure 4) were installed in another large section of the tea plantation, where two rows of tea plants were surrounded by each exclosure. To reduce interaction, the exclosures were set 10 m apart. The frame of the cage was made of aluminum alloy, covered with white nylon net (100 mesh). The cages served to isolate tea plants so that each cage represented an independent small ‘ecosystem’, still sharing the same environmental conditions with the other cages. According to the preliminary measurements, except for the wind speed that was reduced by 50%, the conditions such as temperature, rainfall, relative humidity, and solar radiation inside the cages were not significantly different from outside the exclosures.

The treatments were similar to the field experiment and were randomly assigned to each exclosure. The first treatment was the use of two yellow sticky cards (15 by 20 cm, both sides with glue), which were installed at the center of the cage and between two rows of tea plants at a canopy height (approximately 0.5 m). The cards were replaced every week. For the white, green and yellow light treatments, two modified lamps as described in the field experiment but without the sticky trap (no trapping) were installed between the two rows of tea plants inside a cage, 2 m apart. As previously mentioned, the LED lamps only worked during night-time. The control treatment used the same cages but with no card or lamp. Every week at 7:00 a.m., a suction machine (electromotor, 16,000 rpm, with a 12-cm-diameter inlet) was used to collect and record the number of female and male tea leafhopper adults and spiders in 1 m^2^ area of tea plants in each cage in the five treatments. To minimize the influence of sampling of community dynamics within the exclosures, a mesh bag was installed inside the suction machine to collect all the samples alive (Figure 2). These living samples were quickly identified and recorded, then released back in the cage. Each suction collection lasted 1 min. The experiment began on 25 August and was terminated on 8 December 2015.

### 2.4. Statistical Analyses

#### 2.4.1. Open-Field Experiments

The daily changes in the numbers of adult *E. onukii* (females and males together), parasitoids (all families together) and spiders (all families together), as well as the daily mean female:male ratio of adult *E. onukii* in each treatment, were calculated and graphed. The normality and homogeneity of variance were first tested and were satisfactory for the different dependent variables. First, the block effect was tested as a random effect using an analysis of co-variance. Because the block effect was not significant, it was omitted in further analyses. Analyses of variance were performed to compare the different treatments (excluding the control that had a different sampling method). The dependent variables included the total number of leafhoppers per plot, number of females and males, sex ratio, and numbers of parasitoids and spiders collected in each plot and treatment. Because the graphs detected some important variation in the population dynamics of leafhopper, additional analyses of variance were completed for the May peak, June low, July peak and late-August low, using average daily capture of leafhopper (males or females), parasitoids and spiders. To examine the potential influence of weather conditions (daily temperature, relative humidity, wind speed and solar radiation) on the various capturing techniques, the data were ln-transformed first, to satisfy the normality and homogeneity of variance, then followed with a multiple regression to compare the daily number of leafhoppers, parasitoids and spiders captured by the four traps.

#### 2.4.2. Exclosure Experiment

To satisfy the normality and homogeneity of variance of weekly numbers of leafhoppers and spiders, data were ln-transformed. One-way analysis of variances (ANOVAs) were performed, followed by LSD multiple range tests (*p* ≤ 0.05) to compare the treatments using the total number of leafhoppers and spiders as dependent variables. Repeated measures analyses of variance, using a general linear model, were performed to examine variation over time (weeks) and among treatments (techniques). If the treatment variable was significant, the analysis was followed by LSD multiple range tests (*p* ≤ 0.05) to compare the treatments using the total number of leafhoppers or spiders as dependent variables.

## 3. Results

### 3.1. Sticky Traps Attracted Predators and More E. Onukii Males than Females

The mean total number of leafhoppers collected over the course of the open field trapping experiment ranged from 2272.4 ± 264.3 using the yellow sticky cards to 2616.3 ± 182.1 using the white light at night (Table 1). When the four trapping treatments were compared, the number of females were not significantly different. White light attracted significantly more males than the other three treatments, resulting in the total number also being significant (Table 1).

The daily collection of leafhoppers by suction machine showed that the population size varied over time in a cyclic manner, with two outbreak periods and two low periods (Figure 5a). The first outbreak period (outbreak period A) was 14–28 May, lasting 14 days, and the second outbreak period (outbreak period B) started on July 8 and ended on 25 July, lasting 17 days. There were two low abundance periods: 29 May–7 July (51 days), and 20 August–20 September. The daily numbers of leafhoppers trapped by the four trapping techniques showed a similar trend as the suction machine catch did (Figure 5b–e).

In the four trapping treatments, the mean female to male ratios were all less than 0.3 (white light 0.234 ± 0.035, green light 0.228 ± 0.028, yellow light 0.229 ± 0.013, yellow card 0.214 ± 0.016). The values were significantly lower than the control that had a mean ratio of 0.713 ± 0.014 (Table 1). The mean daily trapped female to male ratios in the four trapping treatments varied more over time than the control. Between July 8 and 12, the female to male ratios were over 0.5 for white, green and yellow lights (Figure 5c–e).

The spiders caught by the suction machine and four different trapping techniques were mainly from the families Araneidae, Oxyopidae, Salticidae, Theridiidae, and Micryphantidae. Among the four trapping treatments, we observed that yellow sticky card trapped the least number of spiders (124 ± 18), though there was no significant difference compared to the others (Table 1). Two outbreak periods (14–28 May and 8–23 July) of spider populations were observed when looking at daily numbers collected by the suction machine (Figure 6a), which corresponded to the population peaks in tea green leafhopper. The four trapping treatments, however, did not show a similar trend (Figure 6b–e).

The trapped parasitoids were from the families Braconidae, Chalcididae, Ichneumonidae and Trichogrammatidae. The mean total number of parasitoids trapped 237 ± 18 by the yellow sticky cards, which was significantly higher than the three light-trapping treatments (Table 1). The daily numbers of parasitoids collected by the suction machine remained relatively constant over time and did not show a peak corresponding to the leafhopper outbreak (Figure 7a). The parasitoids trapped by the four techniques showed slightly higher values in May and during the July outbreak (Figure 7b–e).

### 3.2. Weather Factors Influenced the Effectiveness of The Traps

The potential influence of weather variables on the efficiency of capture was examined by analyzing the variation in the number of leafhoppers trapped as a function of the different weather parameters recorded during the experiment (Table 2). The regression analyses showed that the trapping of leafhoppers by yellow sticky cards was negatively affected by maximum solar radiation (Table 2). Rainfall positively influenced the number of leafhoppers trapped by green and yellow lights. Maximum temperature negatively affected the number of leafhoppers trapped by the suction machine. The number of parasitoids trapped was significantly positively affected by temperatures, with green light related to maximum temperature and yellow with mean temperature. The number of parasitoids trapped by the suction machine was negatively related to minimum temperature. Spiders had the most significant positive relationships with maximum temperature for yellow sticky cards, and minimum temperature for all three lights. This relationship was negative between the number of spiders and maximum temperature for the suction machine (Table 2).

### 3.3. Light without Trap Worked Better than Sticky Cards to Control Leafhopper

According to the daily average number of *E. onukii* females and males during the course of the exclosure experiment (Table 3), white and green light treatments attracted significantly fewer individuals than the other three techniques (*p* < 0.05), with no difference found between yellow sticky card, light treatment and the control. The repeated-measures ANOVAs showed that while the weekly number of tea green leafhoppers significantly varied over time, and even by treatment and time for total number and number of females, there was no significant difference among treatments (Table 4). This was also true for spiders where the weekly numbers varied over time but not among treatments (Table 4). There were two outbreaks of *E. onukii* in all the treatments and control (Figure 8). The outbreak time in the three light treatments was delayed when visually compared to the yellow sticky card treatment and the control (Figure 8). The second outbreak peak happened at about week 9 (20 October) for the yellow sticky card treatment and the control, while at week 11 for the three light treatments (3 November). In weeks 5–8 and 14–16, the numbers of leafhoppers in the green and white light treatments generally were significantly lower values than the other treatments (Figure 8a), according to an exploratory one-way ANOVA for these specific weeks. Female tea green leafhoppers showed similar trends (Figure 8b). Unlike the females, males did not show similar trends, with few significant differences among treatments (Figure 8c). Due to a lack of plant species with nectar in the exclosures, adult parasitoids could not survive. However, the total number of spiders were recorded and showed no significant differences among treatments (Table 3). The weekly number of spiders significantly varied during the 16 weeks (*p* < 0.001), but again no significant effect was found for the treatments (Table 4).

## 4. Discussion

Our experimental results from both open-field and exclosure experiments indicated that light trapping might play a role in leafhopper control. The results from both experiments indicated that light-trapping and even lights alone might be able to contribute to the control of leafhoppers. However, in the open-field experiment, we found that yellow sticky cards and the three types of light traps were strongly attractive to spiders and parasitoids. In tea plantations, spiders and parasitoids play important roles in leafhopper control, and trapping them may have negative impacts on this natural pest control.

Spiders are the main predators of leafhoppers in tea plantation [32]. They hunt nymphs and adults of *E. onukii* for food. On average, one spider can catch ten leafhoppers (nymphs or adults) per day [33]. Both web spiders and non-web spiders (or hunting spiders) feed on tea green leafhoppers. The web spiders prefer to build their webs near artificial lighting where more insects can be preyed on (due to the phototaxis of insects) [34]. It is therefore highly possible that they were trapped on the glue of the lights. Most hunting spiders tend to be daytime hunters and mainly utilize the vibratory signals made by prey for prey-locating [35,36]. However, they still could be trapped when passing by a sticky card.

Several Mymaridae parasitoid species are egg parasitoids of *E. onukii*. In the field, 16–75% of tea leafhopper eggs can be parasitized by them [37]. Unlike the spiders, most adult parasitoids feed on nectariferous plants, which are rarely present within tea plantations [38]. Adult parasitoids usually fly away for food, which could explain that less than half the number of parasitoids were collected by the suction machine, when compared to the four trapping techniques. Adult parasitoids are mainly active and lay eggs during the daytime, and they are attracted by the yellow color, which may explain their largest numbers on the sticky cards [39,40,41].

Lights at night tended to suppress the activities of leafhoppers including moving, cleaning and, most importantly, mating activities, which maybe contributed to a decrease in the population size of *E. onukii* in tea plantations [42]. Previous studies have reported that under controlled conditions, light can change the behaviors of *E. onukii,* including its circadian rhythm [42]. The consequence of this behavioral change may include unsynchronized mating activities between males and females. Such a technique may also be less harmful to the predators than sticky cards, and therefore potentially valuable as a better method for tea green leafhopper control.

Weather conditions could be important factors that influence the effects of sticky traps and light traps in open field. In one way, the stickiness of cards can be affected by rainwater and temperature, and the illumination distance could be decreased by fog (experiences in authors’ experiment). It is also possible that the insects which include the pests and enemies, can adjust their flight, courtship and feeding activities to adapt to changing weather conditions [43]. For example, flying activities of honeybees are negatively influenced by rainfall, humidity, temperature and wind [44,45]. In our opinion, sticky traps and light traps might be more effective in greenhouse pest control than in open field [46,47].

Natural populations of *E. onukii* generally produce equal numbers of males and females [25]. While the suction machine trapped more females than males, the sticky cards tended to attract more males. The higher ratio of females in a population is expected, since female adults tend to live longer than males [25]. Yellow sticky cards and light traps were therefore not as effective if one wants to mainly control the female population. The differences in behavioral activities and spectral sensitivity between male and female may explain these results. Male adults of *E. onukii* are more active than females and their taxis toward yellow color, for example, may increase their likelihood to be trapped [42]. For most insect species, the spectral sensitivity curves of photoreceptor cells are affected by the numbers, types and spatial arrangements of visual pigments, chromophores, screening pigments, and photoreceptor cells in the insect eye, and these variations can be species- or sex-specific [48]. Sexual dimorphism is often linked to adaptive evolution associated with the different activities accomplished by males and females. For example, only *Lycaena* butterfly females possess the visual pigment P568, crucial for the long-range detection (red colors) of larval food plants [49]. This leads females to be more attracted to red than males. Further physiological and genetic studies would be required to better understand the mechanisms underlying the differences between males and females in *E. onukii*.

## 5. Conclusions

The results of our study suggest that light, especially green and white, can be useful as a way to control leafhopper populations without affecting predators too much. Interestingly, green light acts as a monochromatic light, and, more importantly, it is least absorbed by green plants [50], and may not disturb the photosynthesis of tea plants compared to white light. Further research using such a system in operating plantations may help to understand how leafhopper populations can be controlled, while minimising the negative effects on natural pest control agents.

## Figures and Tables

**Figure 1 insects-12-00014-f001:**
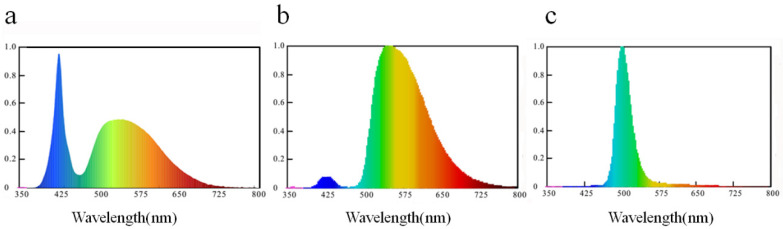
The spectra of the light emitting diode (LED) lights. (**a**) white LED lights; (**b**) green LED lights; (**c**) yellow LED lights.

**Figure 2 insects-12-00014-f002:**
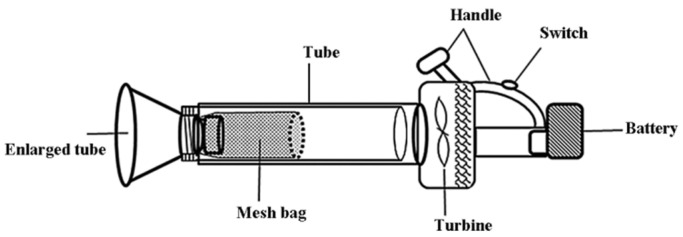
The structure of suction machine.

**Figure 3 insects-12-00014-f003:**
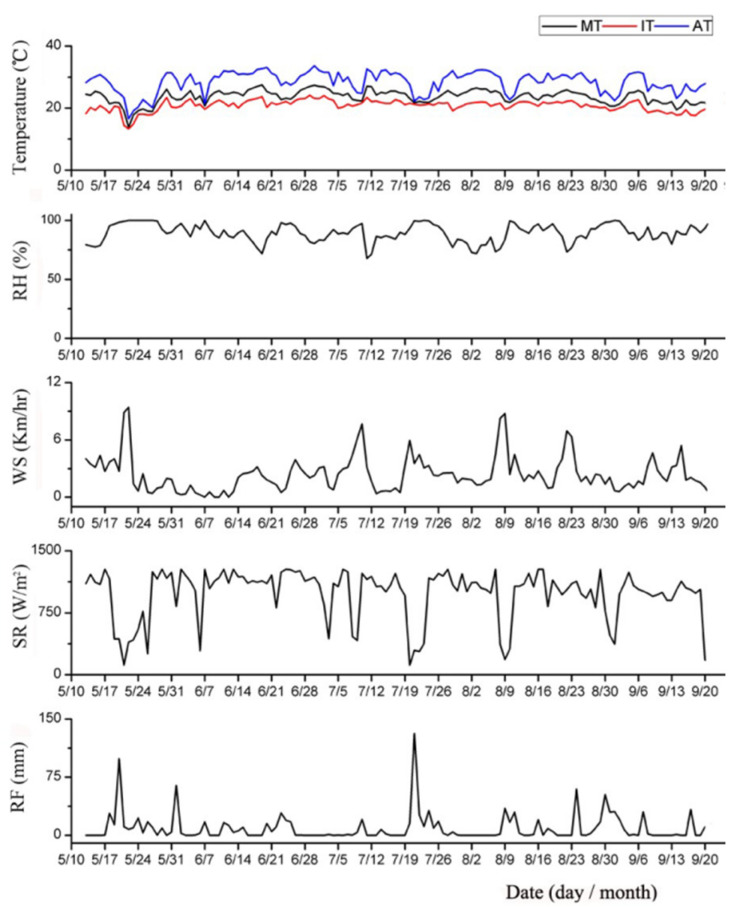
Data of the weather variables during 13 May to 20 September 2015. MT: mean temperature; IT: minimum temperature; AT: maximum temperature.

**Figure 4 insects-12-00014-f004:**
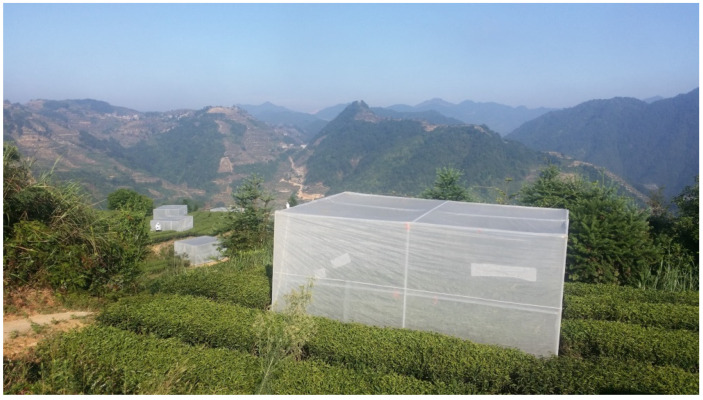
Photo of the cages in the exclosure experiment, Hongxing tea plantation (25°0′15.76″ N, 117°52′0.04″ E; 700–750 m elevation).

**Figure 5 insects-12-00014-f005:**
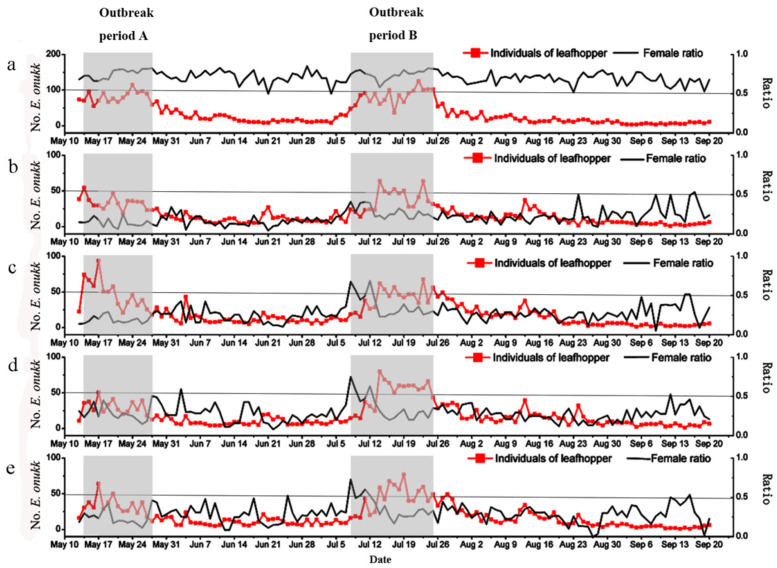
Population dynamics and female ratios of *Empoasca onukii* trapped/collected in the five treatments during the course of field experiment (13 May to 20 September 2015). (**a**) suction machine; (**b**) yellow sticky card; (**c**) white light trap; (**d**) green light trap; (**e**) yellow light trap. The outbreak period A and outbreak period B were covered with gray shades.

**Figure 6 insects-12-00014-f006:**
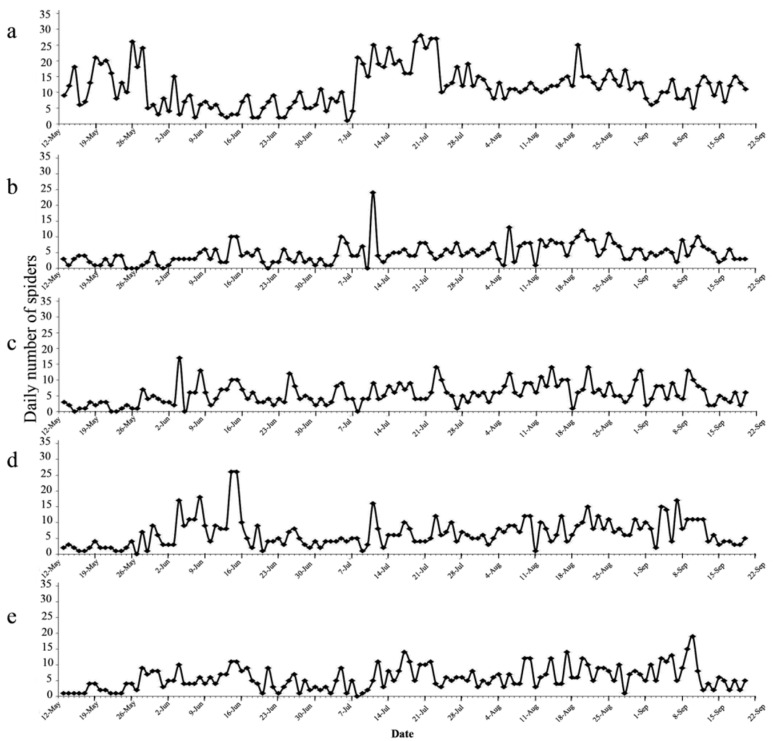
Numbers of spiders trapped/collected in the five treatments during the course of field experiment (13 May to 20 September 2015). (**a**) suction machine; (**b**) yellow sticky trap; (**c**) white light trap; (**d**) green light trap; (**e**) yellow light trap.

**Figure 7 insects-12-00014-f007:**
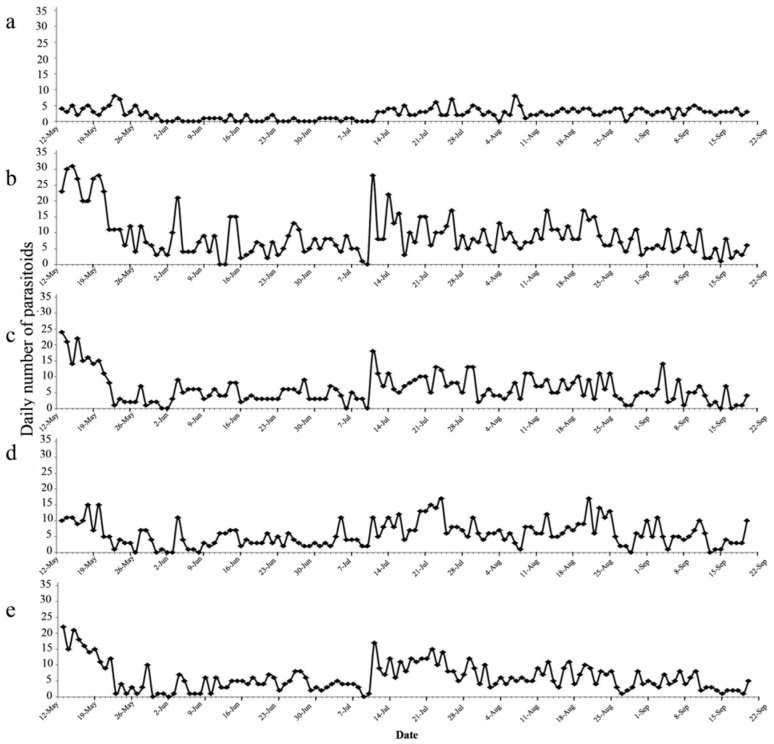
Numbers of parasitoids trapped/collected in the five treatments during the course of field experiment (13 May to 20 September 2015). (**a**) suction machine; (**b**) yellow sticky trap; (**c**) white light trap; (**d**) green light trap; (**e**) yellow light trap.

**Figure 8 insects-12-00014-f008:**
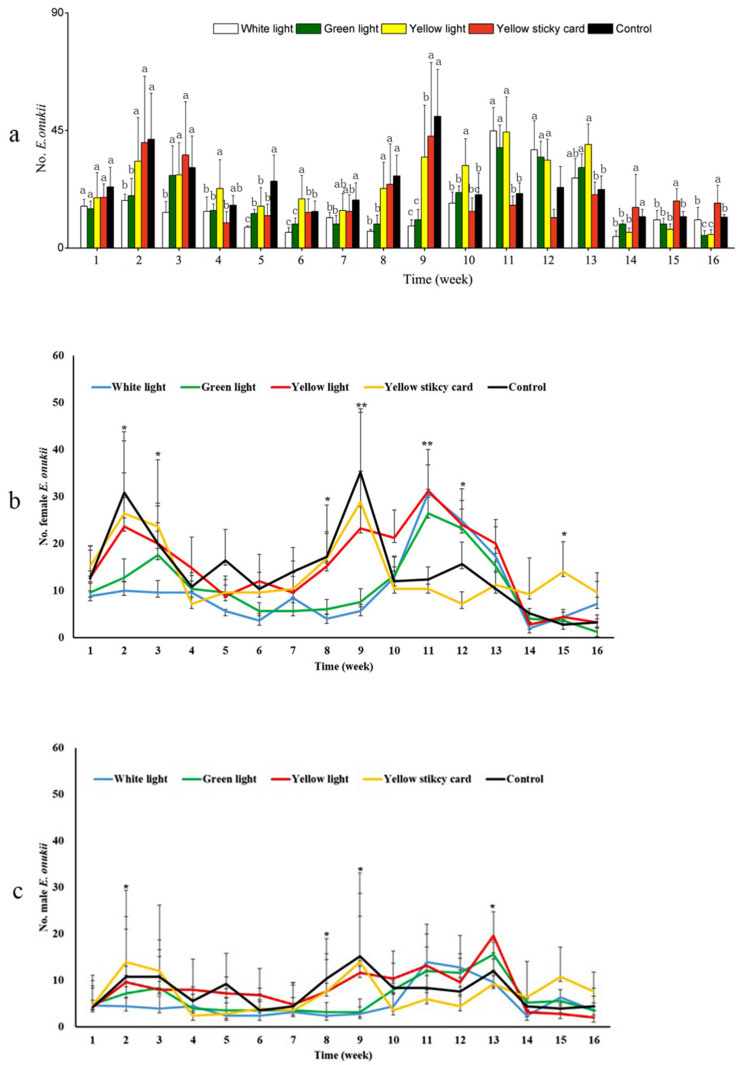
Population dynamics of total *E. onukii* (**a**), female (**b**) and male (**c**) in the five treatments in the exclosure experiment. Different letters within the same week represent significant differences based on a one-way ANOVA followed by an LSD test, *p* < 0.05; Significant differences based on a one-way ANOVA followed by LSD tests, *: *p* < 0.05; **: *p* < 0.001. The weekly collection lasted from 25 August to 8 December, for a total of 16 weeks.

**Table 1 insects-12-00014-t001:** Total number of *Empoasca onukii* and predators and mean female ratio of tea green leafhopper collected during the course of the field experiment.

Technique	Spider	Parasitoid	Female Leafhopper	Male Leafhopper	Total Leafhopper	Female Ratio of Leafhopper
Suction Machine	306.2 ± 13.8	65.4 ± 11.2	1686.4 ± 69.8	604.1 ± 25.3	2290.4 ± 74.5	0.713 ± 0.014
White Light	145.4 ± 14.3 a	160.7 ± 25.2 b	612.4 ± 78.2 a	2003.7 ± 231.1 a	2616.3 ± 182.1 a	0.234 ± 0.035 a
Green Light	173.6 ± 26.9 a	154.4 ± 17.1 b	530.7 ± 68.2 a	1825.4 ± 220.2 b	2354.5 ± 253.3 b	0.228 ± 0.028 a
Yellow Light	151.1 ± 11.7 a	160.4 ± 30.3 b	515.6 ± 66.4 a	1785.3 ± 152.4 b	2301.3 ± 213.7 b	0.229 ± 0.013 a
Yellow Card	124.4 ± 18.2 a	237.1 ± 18.3 a	465.2 ± 60.2 a	1806.5 ± 204.3 b	2272.4 ± 264.3 b	0.214 ± 0.016 a

13 May 2015–20 September 2015. Female ratio = number of female/total leafhoppers. Different letters within the same column of same object represent significant differences based on a one-way analysis of variance (ANOVA) followed by an LSD test, *p* < 0.05. The data of suction machine was not included in above statistical analysis.

**Table 2 insects-12-00014-t002:** Regression equations for the influence of weather variables on the population dynamics of leafhoppers, parasitoids and spiders under the four trapping techniques and the suction machine.

Treatment	Adult *E. Onukii* (*Y*_1_)	Parasitoid (*Y*_2_)	Spider (*Y*_3_)
Yellow sticky card	Ln(*Y*_1_ + 1) = 3.086–0.001*X*_6_, *r* = 0.179, *F* = 4.277, *p* = 0.041	-	Ln(*Y_3_* + 1) = 0.455 + 0.040*X*_3_, *r* = 0.222, *F* = 6.703, *p* = 0.011
White light trap	-	-	Ln(*Y*_3_ + 1) = −0.660 + 0.115*X*_1_, *r* = 0.340, *F* = 16.875, *p* < 0.001
Green light trap	Ln(*Y*_1_ + 1) = 2.598 + 0.009*X*_4_, *r* = 0.221, *F* = 6.649, *p* = 0.011	Ln(*Y*_2_ + 1) = 0.123 + 0.048*X*_3_ + 0.010*X*_4_ + 0.067*X*_7_, *r* = 0.310, *F* = 4.496, *p* = 0.005	Ln(*Y*_3_ + 1) = 0.135 + 0.083*X*_1_, *r* = 0.249, *F* = 8.520, *p* = 0.004
Yellow light trap	Ln(*Y*_1_ + 1) = 2.595 + 0.008*X*_4_, *r* = 0.180, *F* = 4.298, *p* = 0.040	Ln(*Y*_2_ + 1) = −0.545 + 0.085*X*_2_ + 0.007*X*_4_ + 0.097*X*_7_, *r* = 0.364, *F* = 6.486, *p* < 0.001	Ln(*Y*_3_ + 1) = 0.013 + 0.084*X*_1_, *r* = 0.255, *F* = 8.943, *p* = 0.003
Suction machine (population dynamics)	Ln(*Y*_1_ + 1) = 4.295 − 0.061*X*_3_, *r* = 0.242, *F* = 8.029, *p* = 0.005	Ln(*Y*_2_ + 1) = 4.151 − 0.149*X*_1_, *r* = 0.408, *F* = 25.722, *p* < 0.001	Ln(*Y*_3_ + 1) = 3.486 − 0.044*X*_3_ + 0.062*X*_7_, *r* = 0.361, *F* = 9.568, *p* < 0.001

The experiment started on May 13 and ended on 20 September 2015. Stepwise multiple linear regressions were used (*p* < 0.05). Minimum/mean/maximum temperature (X1/X2/X3), rainfall (X4), relative humidity (X5), maximum solar radiation (X6) and wind speed (X7). *r*: correlation coefficient.

**Table 3 insects-12-00014-t003:** Comparison of the daily numbers of tea green leafhoppers and spiders collected during the course of the exclosure experiment.

Object	White Light	Green Light	Yellow Light	Yellow Sticky Card	Control
Total *E. onukii* individuals	11.68 ± 2.22 b	12.61 ± 2.02 b	18.75 ± 2.45 a	17.40 ± 2.35 a	17.04 ± 2.44 a
*E. onukii* female	10.28 ± 1.97 b	10.72 ± 1.77 b	15.43 ± 2.09 a	13.73 ± 1.70 ab	14.3 ± 2.19 a
*E. onukii* male	1.41 ± 0.31 b	1.89 ± 0.35 b	3.32 ±0.48 a	3.68 ± 0.70 a	2.7 ± 0.31 a
Spider	3.21 ± 0.34 a	2.95 ± 0.22 bc	3.05 ± 0.32 ab	2.85 ± 0.23 c	3.20 ± 0.23 a

The daily average of collected tea green leafhopper and spiders of 16 weeks were calculated (mean ± SE). Different letters within the same row of same object represent significant differences, LSD, *p* < 0.05. SE: standard error.

**Table 4 insects-12-00014-t004:** Results of repeated-measures ANOVAs comparing the weekly number of *E. onukii* and spiders collected during the course of the exclosure experiment.

Objective	Source of Variation	Df	*F*-Value	*p*-Value
*E. onukii* (females + males)	Within subject			
	Week	1, 3.917	7.419	<0.001
	Week × treatment	4, 15.668	1.614	0.046
	Between subject			
	Treatment	4, 20	0.361	0.834
Female *E. onukii*	Within subject			
	Week	1, 4.100	8.364	<0.001
	Week × treatment	4, 23.419	1.794	0.035
	Between subject			
	Treatment	4, 20	0.333	0.852
Male *E. onukii*	Within subject			
	Week	1, 5.855	6.052	<0.001
	Week × treatment	4, 16.401	1.356	0.147
	Between subject			
	Treatment	4, 20	0.310	0.868
Spiders	Within subject			
	Week	1, 15	3.585	<0.001
	Week × treatment	4, 60	0.976	0.530
	Between subject			
	Treatment	4, 20	0.762	0.562

Since the sphericity assumption was not satisfied (*p* < 0.05), Huynh–Feldt was used for adjusting.

## Data Availability

The data presented in this study are available on request from the corresponding author.

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
