# Peer review of "Are Yellow Sticky Cards and Light Traps Effective on Tea Green Leafhoppers and Their Predators in Chinese Tea Plantations?"

_insects, 2020, doi:10.3390/insects12010014_

Round 1
Reviewer 1 Report
I enjoyed this study on the effects of light traps on populations of leafhoppers in tea plantations. The authors have designed a complete study, using both open-field and enclosed outdoor experiments and a variety of different trapping and light treatments ,to fully explore their effect on pest populations, as well as beneficial arthropods. Their main findings are interesting, that light alone may be used as a control agent against leafhopper pests, which would prevent accidental trapping of other beneficial arthropods, such as spiders. I suggest minor edits to the text, to improve the clarity of the results section and to include some lacking methodological detail and discussion.
L63: “produces” to “produce”
L65: extra space between “In tea”
L86: “short-wavelength lights” are these UV? Or blue?
L94: this sentence seems unfinished. They authors could put the suction machine in context here e.g. “a suction machine was used, to sample populations over the trial period”
Section 2.2
Are there spectral reflectance data available for the yellow sticky cards, to display the wavelengths of light that are maximally reflected?
Similarly, the authors should provide either spectra for the LED lights used, or provide the manufacturer and model number of each LED. This would give a better insight into the attraction of this species to certain wavelengths.
L111: I am a bit confused by the description of the light trap. Is the sticky card placed inside the lampshade or at one of the openings of the cylinder? If the authors could clarify this description slightly then this would be helpful.
L151: was the 1m2 area sampled, the same each time, or was this rotated each week?
L152: “and graphed” does not make sense here.
L187: It was not clear to me in the text whether the authors were referring to the open field trapping experiment or the exclosure trapping experiment. Some clarification in the text would be useful.
L321 - 322: Is there a reference available for these observations, or were they made by the authors?
L345: The final addition of spectral sensitivity information seems out of place here in the conclusions. The authors might consider placing this at the end of the discussion as it follows from the previous discussion of sex differences between phototaxis and then using the conclusion section to instead summarise their final thoughts. Additionally, the spectral sensitivity of this species has been measured in both males and females, and spectral choices between males and females have also been discussed (see Bian et al., (2020) https://www.mdpi.com/2075-4450/11/7/426).
L356: “is” to “in”, I think?
L356: “as a monochromatic light” sounds strange. Do the authors mean “The interest in green monochromatic light to control insect pest populations, is however more…”?
L359: “while not affecting too much natural pest control agents” to e.g. “while minimising the negative effects on natural pest control agents”.
Reviewer 2 Report
This manuscript was well-written and only requires minor editorial attention. Studies were conducted well and analysis was appropriate. This article contributes nicely to the literature on ecology and management of this important pest species. Suggestion for revision are minor and as floows:
Line 256 fewer not less
Figure 5 might be easier to read if these panels were stacked on each other and took up an entire page
Table 3. what are those means of? They are too low to be the total number of individuals collected from each trap type summed over the course of the study but isnt that what the title suggests? Are they the average per day? In any event, please clarify what these are means of.
Line 354, the experiment is not suggesting, you are using that data to suggest. I would suggest some subtle rewording here.
Line 355. Sentence starting with ‘the interest is….. needs rewording, there seems to be too many “is’s.
Line 359 reword…while not disrupting natural pest control agents.
Reviewer 3 Report
Summary
Authors conducted two experiments to compare the effects of sticky traps and different colors of light on tea leafhopper (TLH) populations in Chinese tea plantations.
Experiment 1 – Trap Comparison
A randomized complete block design was used to compare TLH populations in plots with (1) yellow sticky trap, (2) white light with sticky trap, (3) green light with sticky trap, (4) yellow light with sticky trap and (5) no trap/light control. TLH were counted weekly on the traps and suction sampling was used in the control plot to track population dynamics. This was a 1-year study May – September.
Experiment 2 - Exclosure experiment
Twenty-five cages were established over 2 rows each of tea plants. Five cage each were assigned to treatments (1) yellow sticky trap, (2) white light alone, (3) green light alone, (4) yellow light alone and (5) no trap/light control. Suction sampling was used in each cage to track TLH and natural enemy populations. This was also a 1-year study, August – December.
Results from Experiment 1 found that most trapping methods collected equivalent densities of leafhoppers, with the exception of white light which attracted more males and therefore more overall leafhoppers. Suction sampling (to track population dynamics) tended to collect more females while the sticky traps and lights collected more males, although overall total population counts were similar regardless of the skewed sex ratio. All light and sticky traps also collected a lot of natural enemies though, which may be problematic for biological control.
Results from Experiment 2 found that TLH populations were lower in the cages with the white and green lights, and authors suggest this population effect is due to interference with some aspect of leafhopper behavior. In this experiment parasitoid densities were too low to count, but spider abundance did not appear to be influenced by the different trap/light types.
There was also some response of insects to changes in the weather, and these are noted by the authors.
Authors conclude that white/green lights may be useful to disrupt TLH and lower populations without causing negative impacts on spiders, a key natural enemy in this system.
General Comments
This is an interesting study and authors did a nice job designing and executing the work. My major complaint is that this is only 1 year of field data, and thus insufficient to draw conclusions. I suggest authors collect at least 1 more year of field data before trying to draw any conclusions or submit for publication. It is for this reason that I am rejecting the manuscript.
Overall the manuscript is a bit disorganized, and I provided a number of suggestions to improve clarity and presentation of the data.
My final criticism is minor, but still necessary. The authors claim that reduced TLH populations are due to disruption of TLH behavior by green/white light – maybe so, but at this stage that is rather speculatory, and the way this was phrased in L321-323 seems too certain. So, I suggest authors rephrase this suggestion to say that “maybe” this is what is happening here rather than “certainly”.
Specific Comments
L52: Very unclear what you mean by “not as death traps, but rather as a behavioral control system” – Do you mean not as an attract and kill strategy? What behavior is being controlled by this system?
Was vacuum sampling used in the other plots? Or just in the control plot?
Figure 2: There is some gray shading on the y-axis, unclear why, please clear this up.
Table 2: Very difficult to read this, please reorganize. Suggest just presenting r2, F and P values for each Treatment x Insect combination, as you do in Tables 4-7.
Table 1 + 3: Add spaces between the values, SEM and letter – it is very difficult to read. Do this throughout the entire manuscript, not just these tables.
Figure 5: These figures are very low resolution and difficult to read. Please make them higher resolution and increase font size to improve readability.
Tables 4-6 could be combined into a single table, please do this.
L178 + L254: Please clarify that the data you are analyzing and discussing are the leafhopper counts collected with the suction sampling. It is confusing because in Experiment 1 you are looking at data from the traps themselves, whereas in Experiment 2 you are evaluating the suction data from plants with and without the traps (which is a good approach).
L256: Your suction sampling counts are actually lower in the white/green light treatments, implying that the white/green lights did something to lower leafhopper densities, correct?
L321-323: Where is the justification for this statement?
Edits
L27: Add quotes to ‘hopperburn’ as in L65
L29: tea plantations
L31: enemies remains unclear
L31: with daily weather monitoring
L33 + L40: light traps (cover with sticky cards) that used different wavelengths
L35: results suggested
L43: cards attracted many more
L44: during the course of the open field experiment
L62-63: which can produce 9-17
L65: In tea plantations
L151-152: To minimize and graphed the influence of sampling
L163: daily mean female:male ratio
L301: results from both field and exclosure experiments indicated that
